# A Novel apaQTL-SNP for the Modification of Non-Small-Cell Lung Cancer Susceptibility across Histological Subtypes

**DOI:** 10.3390/cancers14215309

**Published:** 2022-10-28

**Authors:** Anni Qiu, Huiwen Xu, Liping Mao, Buyun Xu, Xiaoyu Fu, Jingwen Cheng, Rongrong Zhao, Zhounan Cheng, Xiaoxuan Liu, Jingsheng Xu, Yan Zhou, Yang Dong, Tian Tian, Guangyu Tian, Minjie Chu

**Affiliations:** 1Department of Epidemiology, School of Public Health, Nantong University, Nantong 226019, China; 2Department of Oncology, Affiliated Nantong Hospital of Shanghai University (The Sixth People’s Hospital of Nantong), Nantong 226001, China; 3Department of Oncology, Jiangdu People’s Hospital of Yangzhou, Yangzhou 225202, China

**Keywords:** alternative polyadenylation, single nucleotide polymorphism, non-small-cell lung cancer, lung adenocarcinoma, susceptibility

## Abstract

**Simple Summary:**

Lung cancer is one of the major public health problems in the world. Genetic variation plays an important role in the development of lung cancer. In this study, we screened apaQTL-SNPs that may influence lung cancer development based on a publicly available database. Then we performed a two-stage case-control population susceptibility study. We found a novel apaQTL-SNP, rs10138506, which might affect lung adenocarcinoma (LUAD) risk by modulating the 3′UTR length of *CHURC1*. At the same time, *CHURC1* could function as a suppressor gene in LUAD with APA regulation. Further experiments indicated that *CHURC1* has two poly(A) sites (proximal and distal) and different genotypes of rs1127968 which in perfect LD with rs10138506 can mediate changes in the lengths of the 3′UTR of the *CHURC1* isoforms by choosing different poly(A) sites. The results of this study may be helpful in providing more genetic basis data for the screening of high-risk populations of lung cancer.

**Abstract:**

Background: Alternative polyadenylation (APA) events may be modulated by single nucleotide polymorphisms (SNPs). Therefore, this study aims to evaluate the association between APA quantitative trait loci (apaQTLs)-related SNPs (apaQTL-SNPs) and non-small-cell lung cancer (NSCLC) risk. Methods: APA-related genes associated with NSCLC (LUAD and LUSC) were first identified, and the respective apaQTL-SNPs of those genes were selected. Then, a two-phase case-control study was performed to evaluate the association between candidate apaQTL-SNPs and NSCLC risk. Results: A total of 7 LUAD- and 21 LUSC-associated apaQTL-SNPs were selected. In the first phase, the apaQTL-SNP rs10138506 was significantly associated with LUAD risk (*p* < 0.05), whereas the other two apaQTL-SNPs (rs1130698 and rs1130719) were significantly associated with LUSC risk (*p* < 0.05). In the second phase, the variant G allele of rs10138506 was still significantly associated with an increased risk of LUAD (OR = 1.42, 95%CI = 1.02–1.98, *p* = 0.038). Functional annotation indicated that the variant G allele of rs10138506 was significantly associated with a higher PDUI value of CHURC1. Meanwhile, 3′RACE experiments verified the presence of two poly(A) sites (proximal and distal) in CHURC1, while qRT-PCR results indicated that different genotypes of rs1127968 which, in perfect LD with rs10138506, can mediate changes in the lengths of the 3′UTR of CHURC1 isoforms. Conclusion: The variant G allele of rs10138506 in CHURC1 was correlated with a longer 3′UTR of CHURC1 mRNA and an increased LUAD risk. Further studies should evaluate the interaction between rs10138506 and different 3′UTR lengths of CHURC1 that regulate LUAD development.

## 1. Introduction

Lung cancer is the second-most frequently diagnosed cancer and is the first leading cause of cancer-related deaths worldwide [1]. Non-small-cell lung cancer (NSCLC) accounts for approximately 85% of all lung cancer cases, with lung adenocarcinoma (LUAD) and lung squamous cell carcinoma (LUSC) as the most common histological types [2]. In recent years, and although the NSCLC risk is much higher among elderly people, its incidence still shows increasing trends in women and younger adults, especially for LUAD [3]. Additionally, the survival rates of patients with NSCLC remain very low; for instance, the 5-year relative survival rate in the world is 26%, which is higher than that in China (<20%) [4,5,6]. Thus, further exploring the NSCLC etiology might help identify high-risk individuals and reduce its incidence in the general population.

NSCLC is associated with multiple environmental factors, with tobacco smoking as one of its main known factors [7]. Genetic factors also play a vital role in NSCLC development [8], and single nucleotide polymorphism (SNP) is the most common type among them [9]. Although genome-wide association studies (GWASs) have identified numerous SNPs associated with NSCLC risk, their biological functions remain to be elucidated if located in non-coding regions [10,11]. Based on the results of the ENCODE project, >80% of the human genome has been reported to be biochemically active, especially outside the well-studied coding regions [12]. Therefore, exploring the underlying biological functions of SNPs in non-coding regions may provide additional insights in elucidating the etiology and molecular genetic mechanisms of NSCLC.

Polyadenylation is an important post-transcriptional regulation mechanism of gene expression [13]. Additionally, alternative polyadenylation (APA) events commonly exist in approximately 70% of human genes [14]. Recently, Xiang et al. systematically explored the roles of APA events in different cancer types and identified that they were frequently detected in NSCLC (both LUAD and LUSC) and other cancers [15]. APA events indicate that pre-mRNA is cleaved at different poly(A) sites by recognizing different polyadenylation signals (PAS), resulting in the production of diverse mRNA isoforms with longer or shorter lengths of 3′untranslated regions (3′UTRs) [16]. Poly(A) sites are mainly located in the 3′UTR and can be divided into distal and proximal regions, i.e., near and far from the 3′end, respectively [17]. Generally, the percentage of distal poly(A) site usage index (PDUI) values are used to quantify APA events and are calculated by dividing the number of transcripts at the distal poly(A) sites based on the total number of transcripts with distal and proximal poly(A) sites. The PDUI value ranges from 0 to 1, and a PDUI close to 1 indicates that the gene tends to use the distal poly(A) site, which may cleave and express mRNA isoforms with a longer 3′UTR. A PDUI of close to 0 represents that this gene tends to utilize the proximal poly(A) site, which may cleave and express mRNA isoforms with a shorter 3′UTR in turn [18].

Increasing evidence has shown that copious oncogenes in multiple cancer types are accompanied by 3′UTR shortening [19], which is also common in the majority of NSCLC-related oncogenes [20]. For example, the oncogene *CSNK1D* commonly uses the proximal poly(A) site in LUAD tumor tissues, resulting in a lower PDUI value (shorter 3′UTR) than that in the adjacent normal lung tissues [15]. Different 3′UTR lengths mediated by APA events could affect mRNA stability, transcription and translation [21]. Thus, it is biologically plausible that APA events may modify the expression of certain corresponding genes by affecting the transcripts with different 3′UTR lengths and further influence LUAD and LUSC development. Further studies exploring the APA event regulators and the underlying regulation mechanisms might provide new clues in elucidating the etiology of LUAD and LUSC.

Recent studies have proposed that some SNPs can act as regulators of APA events [22]. Certain SNPs have been associated with the use of different poly(A) sites, which might cause APA dysfunctions. Furthermore, abnormal APA events might regulate gene expression by generating mRNA isoforms with different 3′UTR lengths. Similar to the eQTL analysis, APA quantitative trait loci (apaQTL) analysis can indicate the relationship between different SNP genotypes and PDUI values in certain genes. The proposed novel apaQTL mechanism may provide a novel insight to the interpretation of potential SNP functions in non-coding regions [23]. Recently, by integrating the SNP genotype and APA data (quantified by PDUI values), Yang et al. systematically identified apaQTL-SNPs in 32 cancer types and developed an online database known as SNP2APA [24]. However, apaQTL-SNPs from SNP2APA were obtained by bioinformatics analysis. Further studies with large samples are needed to investigate whether and how apaQTL-SNPs identified by SNP2APA affect NSCLC risk.

In this study, APA-related genes associated with NSCLC (LUAD and LUSC) were firstly obtained from published data by Xiang et al. [15]. Secondly, the respective apaQTL-SNPs for LUAD and LUSC were systematically screened based on these genes. Thirdly, a two-phase case-control study (phase I: 4107 patients with NSCLC and 3710 healthy controls; phase II: 779 patients with NSCLC and 667 healthy controls) was performed to evaluate the association between candidate apaQTL-SNPs and NSCLC (LUAD and LUSC) susceptibility.

## 2. Materials and Methods

### 2.1. Study Population

To explore the association between candidate apaQTL-SNPs and NSCLC risk, a two-phase case-control study (phase I and II) was performed. The study participants in phase I were obtained based on the Female Lung Cancer Consortium in Asia (FLCCA) GWAS from the database of Genotypes and Phenotypes (dbGAP), with accession number phs000716.v1.p1. The FLCCA GWAS consisted of 5510 never-smoking female patients with lung cancer and 4544 controls [25]. Among them, 4107 with NSCLC (including 3453 with LUAD and 654 with LUSC) and 3710 controls obtained from 14 studies conducted in mainland China, South Korea, Japan, Singapore, as well as Taiwan, China and Hong Kong, China were available for the present study.

In phase II, 779 patients with NSCLC (576 LUAD and 203 LUSC) and 667 cancer-free healthy controls were included to further validate the association between candidate apaQTL-SNPs and NSCLC risk. Among the samples in this phase, 626 patients with NSCLC and 667 healthy controls were previously described [26]. An additional 153 patients, newly diagnosed with NSCLC, were continuously recruited from two hospitals (the Sixth People’s Hospital of Nantong and Jiangdu People’s Hospital of Yangzhou) in Jiangsu from 2020 to 2022. All patients were pathologically confirmed. Written informed consent was obtained from all subjects, and the study was reviewed and approved by the ethics committee of Nantong University (approval no. 2022-2, February 2022).

### 2.2. Identification of APA-Related Genes Associated with LUAD and LUSC

APA-related genes associated with LUAD and LUSC were obtained based on the published study by Xiang et al. [15], in which the authors first downloaded the RNA-seq data from The Cancer Genome Atlas (TCGA) database and used the DaPars algorithm to calculate PDUI values of the corresponding genes for quantifying APA events. They further systematically assessed the correlation between PDUI values and expression levels of the corresponding genes using Spearman’s correlation (Rs), and APA-related genes were defined based on the threshold of |Rs| >0.3 and a false discovery rate (FDR) of <0.05.

### 2.3. Selection of Respective apaQTL-SNPs in APA-Related LUAD and LUSC Genes

Respective apaQTL-SNPs in APA-related LUAD and LUSC genes were downloaded from the SNP2APA online website (http://gong_lab.hzau.edu.cn/SNP2APA/ accessed on 21 June 2021) [24], and the authors obtained the SNP genotype data from TCGA database, whereas PDUI values of the corresponding genes were obtained from TC3A database. Then, linear regression of MatrixEQTL was used to calculate the absolute value of the correlation coefficient (r) to evaluate the pairwise association between SNPs and PDUI values. Moreover, SNPs with |r| ≥0.3 and *P*_FDR_ < 0.05 were defined as apaQTL-SNPs.

After obtaining apaQTL-SNPs in APA-related LUAD and LUSC genes, the candidate apaQTL-SNPs were then filtered out with a minor allele frequency (MAF) of ≥0.01 in the East Asian population (including the Chinese Han population, Southern Han Chinese population and Japanese population). Finally, the linkage disequilibrium (LD) analysis with an *r*^2^ threshold of 0.80 was performed to further filter above the apaQTL-SNPs (Figure 1).

### 2.4. Genotyping of FLCCA GWAS in Phase I

In phase I, genotyping of FLCCA GWAS was performed using Illumina SNP microarrays.

### 2.5. Genotyping of Candidate apaQTL-SNPs in Phase II

For the samples in phase II, genomic DNA was extracted from the peripheral blood using a DNA extraction kit (Qiagen, Valencia, CA, USA). Then, genotyping was performed using the TaqMan allele discrimination method, and the Applied Biosystems™ QuantStudio™ 5 FAST Real-Time Polymerase Chain Reaction (PCR) system (Applied Biosystems, Foster City, CA, USA) was used to perform PCR reactions and read the fluorescent signal. Genotyping results were displayed on the allelic discrimination system based on FAM and VIC fluorescence intensity.

### 2.6. 3′-Rapid Amplification of cDNA Ends (3′RACE) Experiments

Total RNA was extracted from A549 cell lines using a TRIzol reagent (Invitrogen). Furthermore, the first-strand cDNA was synthesized from 1 μg of the total RNA using the HiScript 1st Strand cDNA Synthesis Kit (Vazyme R111-02). Then, 3′RACE was used to determine poly(A) sites of Churchill Domain Containing 1 *(CHURC1)* with a 3′RACE kit (Jingrui, Guangzhou, China). The sequences of gene-specific forward primers (GSP) were designed for *CHURC1* as follows:

*CHURC1*_F1: GGTAATACCTGCCTGGAGAATGGATC

*CHURC1*_F2: ACAAGGATCTGGCAAGGTTAGGAAG

PCR products of 3′RACE were then separated through agarose gel electrophoresis and gel-purified. After confirming the quality with agarose gel electrophoresis, the 3′ ends were sequenced using an Applied Biosystems 3730× l DNA Analyzer (Applied Biosystems, Foster City, CA, USA). Finally, the AlignX function of Vector NTI Advance 11.0 (Invitrogen, Carlsbad, CA, USA) was used for sequence alignment.

### 2.7. Vectors Construction and qRT-PCR Assay

The psiCHECK2-*CHURC1*-3′UTR-rs1127968-wid and the psiCHECK2-*CHURC1*-3′UTR-rs1127968-mut were constructed by NKY GENEREADER (Guangzhou, China). Then, qRT-PCR assays were performed to evaluate the relative expression levels of the longer and shorter 3′UTRs of *CHURC1* under rs1127968 wid/mut genotypes. The qRT-PCR was conducted using a SYBR Premix Ex Taq kit (Takara) on a 7500 Real-Time PCR system (Applied Biosystems, Foster City, CA, USA). The sequences of the target longer and shorter 3′UTRs of *CHURC1* were as follows:

*CHURC1*-3′UTR-short-F CCCCCGACAAATGACTCTCTTA

*CHURC1*-3′UTR-short-R AACCATCAGGCTTGCTGTATTG

*CHURC1*-3′UTR-long-F TGCTGCAAATCTATATCTGACC

*CHURC1*-3′UTR-long-R TTAGCAAACTCAAGGAAGGGAC

### 2.8. Statistical Analysis

The two-sided chi-squared test and Student’s *t*-test were used to statistically evaluate differences in demographic characteristics and smoking status between the case and the control groups at each study phase. Logistic regression analyses were used to assess the association between the candidate apaQTL-SNPs and NSCLC risk, and the odds ratios (ORs) and 95% confidence intervals (95% CIs) were obtained after adjusting for gender, age and smoking status as appropriate. Kaplan–Meier survival curves were plotted using the “survival” package in R software. A log-rank test was used to compare the difference in survival rate between the two groups. The “Coxph” function in the “survival” package was used for univariate COX regression analysis. *p* < 0.05 was defined as the boundary of statistical significance. Analysis were performed using Stata version 15.0 (StataCorp, College Station, TX, USA), SPSS version 20.0 (IBM Corp., Armonk, NY, USA) and R version 3.6.2 software (R Foundation for Statistical Computing, Vienna, Austria).

## 3. Results

### 3.1. Characteristics of the Study Subjects

In the present study, the baseline characteristics of patients with NSCLC and cancer-free healthy controls are shown in Table 1. Briefly, a total of 4107 patients with NSCLC (3453 with LUAD and 654 with LUSC) and 3710 healthy controls were enrolled at phase I, and 779 patients with NSCLC (576 with LUAD and 203 with LUSC) and 667 healthy controls were enrolled at phase II. In phase II, the 779 NSCLC cases (including 511 males and 268 females) and 667 healthy controls (including 443 males and 224 females) were used to validate the results. As shown in Table 1, in phase I, 49.38% of patients in the case group were aged ≤60 years, making them younger than those in the control group, and 50.62% of patients in the case group were aged >60 years, making them older than those in the control group. In phase II, the age of patients with NSCLC was comparable to that of the control group (*p* = 0.358), and the gender ratio was comparable between patients with NSCLC and the controls (*p* = 0.781). Additionally, the proportion of smokers in the case group was higher than that in the control group in phase II (*p* < 0.001).

### 3.2. Identification of APA-Related Genes Associated with LUAD and LUSC

In published studies [15], 518 APA-related genes associated with LUAD, and 489 APA-related genes associated with LUSC, were obtained. Among the 518 APA-related genes associated with LUAD, the PDUI values of 233 genes were negatively correlated with the corresponding gene expression (R_S_ < −0.3, *P*_FDR_ < 0.05), and the PDUI values of 285 genes were positively correlated with the corresponding gene expression (R_S_ > 0.3, *P*_FDR_ < 0.05). Among the 489 APA-related genes associated with LUSC, 259 genes had PDUI values negatively correlated with the corresponding gene expression (R_S_ < −0.3, *P*_FDR_ < 0.05), and 230 genes had PDUI values positively correlated with (R_S_ > 0.3, *P*_FDR_ < 0.05).

### 3.3. Screening of Candidate apaQTL-SNPs

Firstly, the apaQTL-SNPs were obtained from the SNP2APA database (16,886 apaQTL-SNPs associated with LUAD and 18,913 apaQTL-SNPs associated with LUSC). Secondly, among these apaQTL-SNPs, 49 apaQTL-SNPs were located in 7 APA-related genes for LUAD, and 137 apaQTL-SNPs were located in 17 APA-related genes for LUSC. Thirdly, based on the MAF value of ≥0.01 in both the Chinese Han and Japanese population, 48 LUAD-related and 114 LUSC-related apaQTL-SNPs were selected. Finally, linkage disequilibrium (LD) analysis (*r*^2^
*<* 0.8) was performed to filter apaQTL-SNPs. Finally, 28 apaQTL-SNPs (7 apaQTL-SNPs associated with LUAD and 21 apaQTL-SNPs with LUSC) were selected (Figure 1, Table 2).

### 3.4. Association between Candidate apaQTL-SNPs and NSCLC Risk in Phase I

In the first phase, the association between 28 apaQTL-SNPs and NSCLC risk was evaluated in the FLCCA GWAS database. Among the 28 apaQTL-SNPs, the SNP rs10138506 located in *CHURC1* was significantly associated with an increased LUAD risk (additive model: OR = 1.16, 95% CI = 1.01–1.33, *p* = 0.034). However, this SNP was not associated with LUSC risk (*p* = 0.325). SNPs rs1130698 (additive model: OR = 1.19, 95% CI = 1.00–1.41, *p* = 0.049) and rs1130719 (additive model: OR = 1.26, 95% CI = 1.08–1.48, *p* = 0.002) located in *CD151* were significantly associated with an increased LUSC risk. However, no statistical significance was observed between the above two apaQTL-SNPs and LUAD risk (both *p* > 0.05) (Table 3).

### 3.5. Evaluation of the Association between apaQTL-SNP (rs10138506) and LUAD Risk in Phase II

To further evaluate the association between the identified apaQTL-SNP (rs10138506) and LUAD risk, genotyping was performed on 576 patients with LUAD and 667 healthy controls from the case-control study of phase II. As shown in Table 4, the variant G allele of rs10138506 was significantly associated with an increased LUAD risk (additive model: OR = 1.42, 95% CI = 1.02–1.98, *p* = 0.038).

### 3.6. Evaluation of the Association between apaQTL-SNPs (rs1130698 and rs1130719) and LUSC Risk in Phase II

To further evaluate the association between the identified apaQTL-SNPs (rs1130698 and rs1130719) and LUSC risk, genotyping was performed on 203 patients with LUSC and 667 healthy controls from the case-control study in phase II. However, no statistical significance was observed between the two apaQTL-SNPs and LUSC risk (both *p* > 0.05) (Table 4).

### 3.7. APA Analysis for rs10138506

We further evaluated the relationship between the identified apaQTL-SNP rs10138506 and PDUI values of *CHURC1* in LUAD. As shown in Figure 2, the apaQTL-SNP rs10138506 was significantly correlated with the PDUI values of *CHURC1* (*p* = 3.33 × 10^−14^). In detail, compared with those of the wild-type homozygote AA genotype, the PDUI values of the heterozygous AG genotype in rs10138506 were significantly higher, whereas those of the variant homozygous GG genotype were even higher.

### 3.8. CHURC1 Expression

Based on TCGA database, the *CHURC1* expression between LUAD tumor tissues (n = 526) and adjacent non-tumor tissues (n = 108) was significantly different (*p <* 0.001), and lower *CHURC1* expression levels were found in tumor tissues. Furthermore, *CHURC1* expression levels in 57 LUAD paired samples were also analyzed, and the results showed that *CHURC1* expression in LUAD tumor tissues was significantly lower than that in the adjacent non-tumor tissues (*p* = 0.002) (Figure 3A,B).

### 3.9. Survival Analysis

To explore whether the expression and PDUI value of *CHURC1* was related to the survival rate, we downloaded the TCGA dataset and the APA-related genes dataset [15] to perform survival analysis. The results showed that the survival time of high *CHURC1* expression in LUAD was significantly higher than that of low *CHURC1* expression (Log-rank *p* = 0.049). (Figure 4A). Further analysis of the correlation between the PDUI value of *CHURC1* and survival time showed that the survival time of the low PDUI value of *CHURC1* in LUAD was higher than that of the high PDUI value (log-rank *p* = 0.056) (Figure 4B).

### 3.10. Analysis of Poly(A) Sites of CHURC1 Based on the 3′RACE Experiment

Bioinformatic analysis indicated that different genotypes of apaQTL-SNP rs10138506 may affect the usage of different poly(A) sites in *CHURC1*, resulting in changes in PDUI values. Then, the predicted poly(A) sites of *CHURC1* were identified in the NCBI database (https://www.ncbi.nlm.nih.gov/ accessed on 16 February 2022), and *CHURC1* 3′RACE was performed and verified using the mRNA from the A549 cell line. These results indicated that *CHURC1* has two poly(A) sites that may produce two mRNA transcripts with different 3′UTR lengths. The distance between the proximal and distal poly(A) sites was approximately 2.7 kb. Meanwhile, SNP rs1127968 was found with a perfect high LD with rs10138506 (*r*^2^ = 1), which was located in the 3′UTR of *CHURC1* and was approximately 0.2 kb away from the poly(A) site1 (Figure 5A). Meanwhile, the APA regulation for rs1127968 was the same as that for rs10138506 (Figure 5B). For the shorter *CHURC1* mRNA isoform, the adopted PAS signal AAUACA is located at 39 bp upstream of the poly(A) site1, and the band of the PCR product (~420 bp) was consistent with the designed sequences (Figure 5C). For the longer *CHURC1* mRNA isoform, the adopted PAS signal UUUAAA was located at 115 bp upstream of the poly(A) site2, whereas the band of the PCR product (~560 bp) was consistent with the designed sequences (Figure 5D). Finally, PCR product sequences were aligned to the transcript reference sequences from the NCBI database. Results showed that the 3′UTR transcript sequences with different *CHURC1* lengths matched with the target region.

### 3.11. The Expression of CHURC1 Isoforms under Different Genotypes of apaQTL-SNP rs10138506

We then performed qRT-PCR to evaluate whether different genotypes of apaQTL-SNP rs10138506 may affect the expression of different lengths of *CHURC1* 3′UTR isoforms. As rs1127968 had a perfect high LD (*r*^2^ = 1) with rs10138506 and was approximately 0.2 kb away from the poly(A) site1, we further explored the association between the different genotypes of rs1127968 and the expression of *CHURC1* isoforms. The results showed that, with the expression level of the shorter 3′UTR of the *CHURC1* mRNA isoform as a reference compared with the rs1127968 wild G allele, the expression level of the longer 3′UTR of the *CHURC1* mRNA isoform was 1.85 fold higher under the rs1127968 mutant C allele. This indicates that different genotypes of rs1127968 may mediate changes in the lengths of the 3′UTR of the *CHURC1* isoforms.

## 4. Discussion

Based on the present two-phase case-control study, the relationship between apaQTL-SNPs and NSCLC risk was systematically appraised. These results showed that the variant G allele of apaQTL-SNP rs10138506 in *CHURC1* was significantly associated with an increased LUAD risk. Furthermore, the variant G allele of rs10138506 was significantly associated with higher PDUI values of *CHURC1*. This study may shed light on the etiology and underlying molecular genetic mechanisms of LUAD.

*CHURC1*, a transcriptional activator controlling the fibroblast growth factor [27], is a cancer susceptibility gene, affecting cell proliferation [28]. In our present study, the *CHURC1* expression level in LUAD tumor tissues was lower than that in the adjacent non-tumor tissues based on TCGA data, indicating that *CHURC1* may function as a tumor-suppressor gene in LUAD development. Furthermore, transcription factor *SIN3A* may bind at the *CHURC1* promoter in human A549 LUAD cell lines [29]. Additionally, reduced *SIN3A* might strongly promote A549 cell invasion [30]. Therefore, *CHURC1* may exert tumor-suppressive effects through certain transcription factors (e.g., *SIN3A*) and further affect LUAD development. We also analyzed the protein expression of *CHURC1* in LUAD based on a published database [31]. The results show that *CHURC1* had no statistically significant expression difference between LUAD tumor tissues and adjacent non-tumor tissues (*p* = 0.33). Although the mRNA expression of *CHURC1* between LUAD tumor tissues and adjacent non-tumor tissues was significantly different (*p* < 0.001), the protein expression of *CHURC1* had no statistically significant difference. Considering the concordance rate of the differential expression of mRNAs and proteins in tissues is relatively low, only about 40% [32], in subsequent studies, screening the genes related to LUAD from the perspectives of the consistency of both mRNA and protein levels are warranted.

At present, relevant studies have shown that the risk factors for lung cancer include exposure to tobacco smoke [33], indoor air pollution caused by cooking smoke and coal combustion [34], outdoor air pollution (PM2.5, et al.) [35], infection and genetic factors [36]. Among them, although environmental risk factors (such as smoking and indoor and outdoor air pollution) have a great impact on the risk of lung cancer, genetic factors explain about 20% of the heritability of lung cancer [37]. In terms of genetic factors, many scholars have explored the impact of genetic variation on LUAD. Dai et al. found that 19 susceptible loci were significantly associated with the risk of non-small-cell lung cancer through a large prospective cohort study of Chinese people, including two new SNP loci for LUAD and one new SNP loci for LUSC [8]. The risk loci found in the study explained the genetic basis of lung cancer, and people with a high genetic risk have a significantly higher risk of lung cancer than people with a low genetic risk.

Studies have found that some tumor-suppressor genes tend to produce mRNAs with longer 3′UTRs in tumor tissues. Begik et al. performed an APA analysis of gene expression data from various cancer types and found that *PNRC1* acts as a tumor-suppressor gene, which tends to produce mRNAs with longer 3′UTRs in tumor tissues [38]. Additionally, *ZBTB4* is significantly downregulated in breast cancer tissues with longer 3′UTRs [39]. Therefore, the role of identified apaQTL-SNP rs10138506 in regulating the 3′UTR length was explored by mediating the *CHURC1* expression. Furthermore, SNP2APA data results showed that the PDUI value of *CHURC1* increased under the variant G allele of rs10138506. Meanwhile, 3′RACE experiments verified the presence of two poly(A) sites (proximal and distal) in *CHURC1*, and the SNP rs1127968 located 0.2 kb downstream of APA site1 was in a perfect high LD with rs10138506 (*r*^2^ = 1.0). Additionally, qRT-PCR assay indicated that *CHURC1* tends to use the distal poly(A) site under the variant C allele of rs1127968, which may express transcripts with longer 3′UTRs, and the potential mechanism underlying the alteration of 3′UTR length that regulates *CHURC1* expression levels remains to be further explored.

The 3′UTR contains various regulatory elements, including miRNA binding sites, which play a significant role in the expression level and translation ability of their target genes. Changes in the 3′UTR length of mRNAs mediated by APA events may affect the existence of miRNA binding sites, which may in turn affect the target genes [40,41]. Several potentially conserved binding sites in the 3′UTR of *CHURC1* were further identified using Targetscan (http://www.targetscan.org/vert_71/ accessed on 18 April 2022). The results showed that both miR-4262 and miR-181b-5p were identified as potential *CHURC1* regulators. Previous studies reported that miR-181b-5p may play a regulatory role in LUAD, which was down-regulated in LUAD [42]. Furthermore, miR-4262 was significantly down-regulated in gastric cancer tissues, and its lower-level expression was associated with a poorer prognosis and lower survival rate in patients with gastric cancer [43]. Taken together, we hypothesized that the variant G allele of rs10138506 could influence the APA events through its high LD SNP rs1127968, thereby making the 3′UTR expression longer in *CHURC1* mRNA. Changes in the 3′UTR length of mRNA yield the enrichment of miRNA binding in the 3′UTR of *CHURC1* and may, therefore, inhibit the *CHURC1* expression in LUAD and elevate the risk of developing LUAD (Figure 6).

This study has several merits. Firstly, the well-reported NSCLC-related APA genes were integrated with the SNP2APA apaQTL-SNP database, which may contribute to the precise screening of candidate NSCLC-related apaQTL-SNPs. Secondly, although LUAD and LUSC are the two main histological NSCLC subtypes, they not only have different histological origins, but also present different molecular oncogenic mechanisms. Thus, apaQTL-SNPs associated with both LUAD and LUSC were separately screened, and the association between screened apaQTL-SNPs and the risk of two different NSCLC pathological types were evaluated. In addition, the conclusion that apaQTL-SNP rs10138506 may modify NSCLC susceptibility across histological subtypes has enhanced the rigor of our study strategy. Thirdly, a two-phase case–control study with a large sample size was conducted to evaluate the association between apaQTL-SNPs and NSCLC risk, which may increase the stability and reliability of the results. The results of this study may be helpful in providing more genetic basis data for the screening of high-risk populations of lung cancer, and as an important reference in establishing a feasible lung-cancer-screening program for precision prevention and individualized intervention. However, certain limitations unavoidably exist in this study. The samples in the first phase of GWAS screening were all females. Although samples in the second phase of the validation included both males and females, we cannot exclude the possibility that gender bias in the first phase may have affected our results, and further studies with strict inclusion criteria should be paid attention.

## 5. Conclusions

This study concluded that the variant G allele of the apaQTL-SNP rs10138506 in *CHURC1* could correlate with longer 3′UTR lengths of *CHURC1* mRNA and increase LUAD risk. Therefore, further investigation is warranted to explore the potential biological effects of rs10138506 and *CHURC1* in LUAD development.

## Figures and Tables

**Figure 1 cancers-14-05309-f001:**
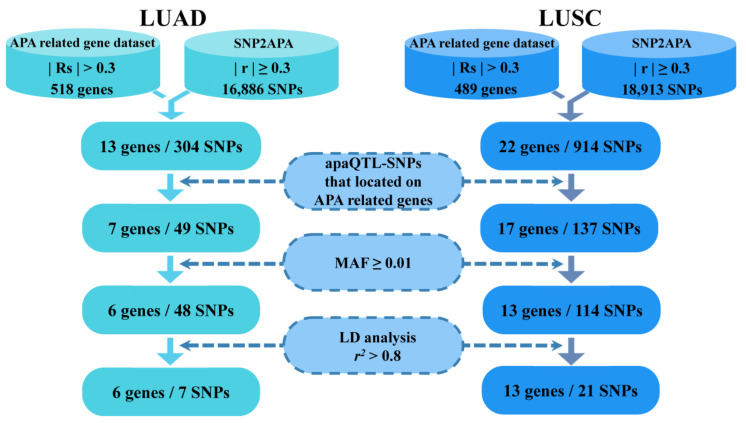
The selection of functional apaQTL-SNPs. apaQTL, alternative polyadenylation quantitative trait loci; SNP, single nucleotide polymorphisms; LUAD, lung adenocarcinoma; LUSC, lung squamous cell carcinoma; RS, Spearman’s correlation; r, correlation coefficient; MAF, minor allele frequency; LD, linkage disequilibrium.

**Figure 2 cancers-14-05309-f002:**
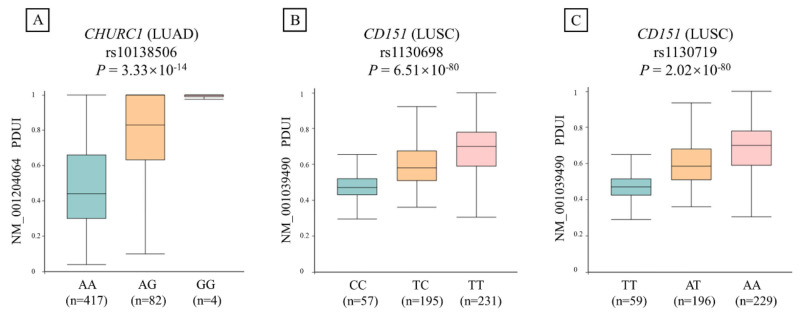
The relationship between three candidate apaQTL-SNPs and PDUI values of their target genes in NSCLC. (**A**) The variant G allele of apaQTL-SNP rs10138506 was significantly associated with higher PDUI values in LUAD. (**B**) The variant C allele of apaQTL-SNP rs1130698 was significantly associated with lower PDUI values in LUSC. (**C**) The variant T allele of apaQTL-SNP rs1130719 was significantly associated with lower PDUI values in LUSC. apaQTL, alternative polyadenylation quantitative trait loci; SNP, single nucleotide polymorphisms; PDUI, percentage of distal poly(A) site usage index; NSCLC, non-small-cell lung cancer; LUAD, lung adenocarcinoma; LUSC, lung squamous cell carcinoma.

**Figure 3 cancers-14-05309-f003:**
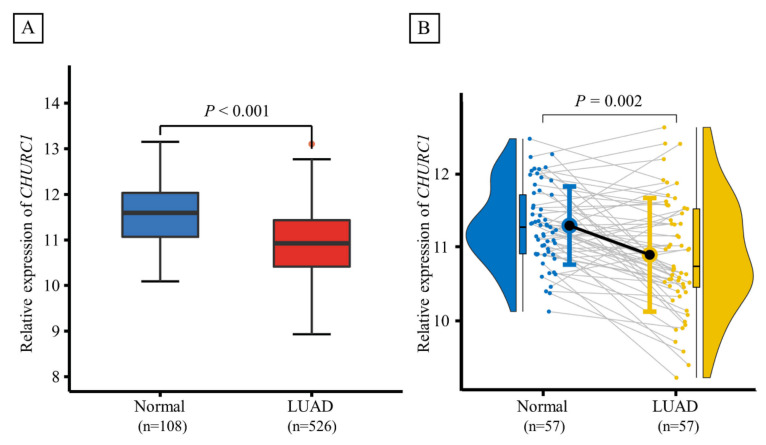
Expression levels of *CHURC1* in LUAD tumor tissues and adjacent non-tumor tissues. (**A**) *CHURC1* expression was lower in LUAD tumor tissues (n = 108) than in adjacent non-tumor tissues (n = 526). (**B**) In 57 LUAD paired samples, *CHURC1* expression was lower in LUAD tumor tissues than in adjacent non-tumor tissues. LUAD, lung adenocarcinoma.

**Figure 4 cancers-14-05309-f004:**
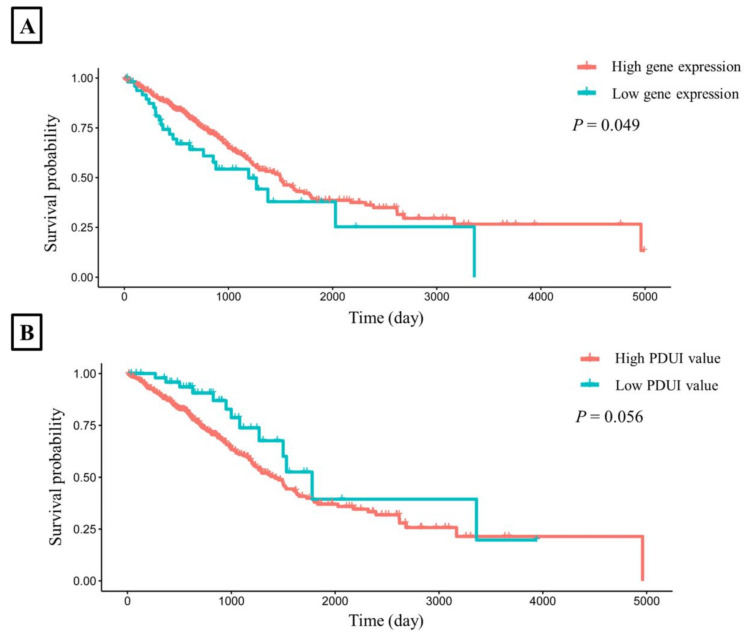
The LUAD survival time curves for *CHURC1*. (**A**) Survival analysis of *CHURC1* expression. (**B**) Survival analysis of PDUI value of *CHURC1*. LUAD, lung adenocarcinoma; PDUI, percentage of distal poly(A) site usage index.

**Figure 5 cancers-14-05309-f005:**
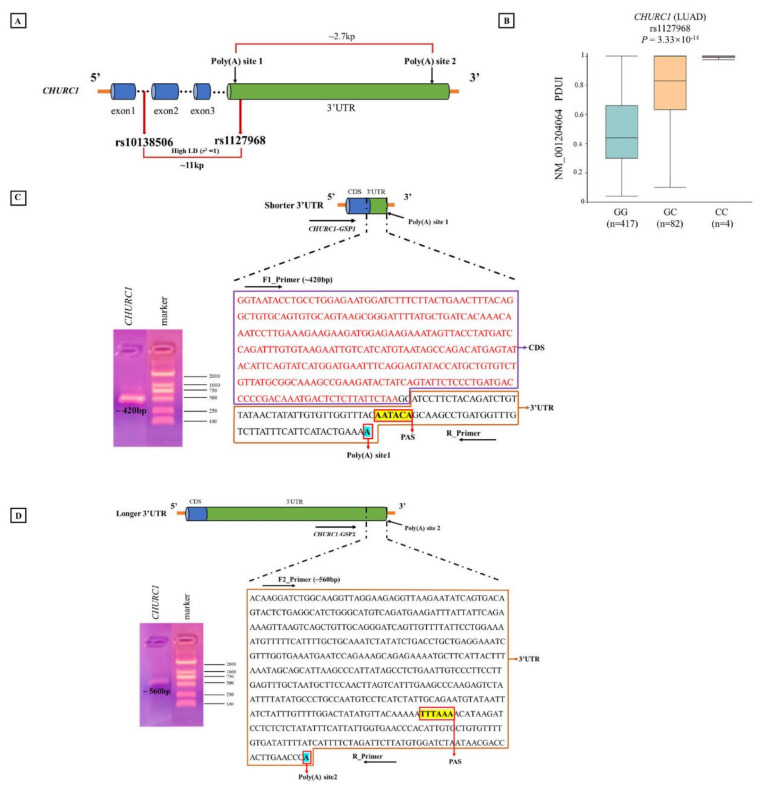
3′RACE experiment determining poly(A) sites of *CHURC1*. (**A**) The location of two poly(A) sites in *CHURC1*; (**B**) the relationship between apaQTL-SNP rs1127968 and PDUI values of *CHURC1* in LUAD; (**C**) the schematic diagram of shorter 3′UTR of *CHURC1* transcript and the sequence of the PCR product (~420 bp); (**D**) the schematic diagram of longer 3′UTR in the *CHURC1* transcript and sequence of the PCR product (~560 bp). 3′RACE, 3′-rapid amplification of cDNA ends; LD, linkage disequilibrium; apaQTL, alternative polyadenylation quantitative trait loci; SNP, single nucleotide polymorphisms; PDUI, percentage of distal poly(A) site usage index; 3′UTR, 3′untranslated region; LUAD, lung adenocarcinoma; CDS, coding sequence; PAS, polyadenylation signal; GSP, gene-specific forward primer.

**Figure 6 cancers-14-05309-f006:**
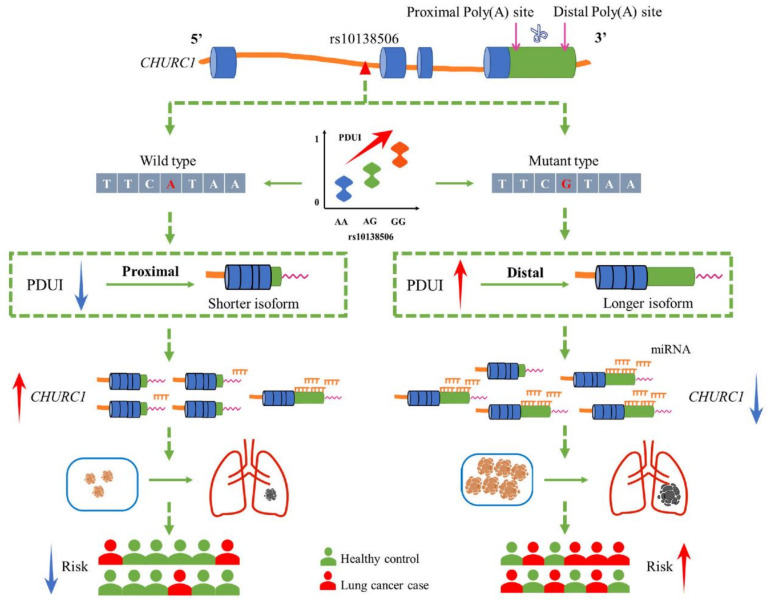
The potential regulation mechanism of apaQTL-SNP rs10138506 mediating malignant phenotypic changes in LUAD by regulating the 3′UTR length of *CHURC1*. apaQTL, alternative polyadenylation quantitative trait loci; SNP, single nucleotide polymorphisms; 3′UTR, 3′untranslated region; LUAD, lung adenocarcinoma; PDUI, percentage of distal poly(A) site usage index; miRNA, mircroRNA.

**Table 1 cancers-14-05309-t001:** Characteristics of the subjects enrolled in this study.

Variables	Screening (FLCCA GWAS)			Validation (Taqman)		
	Case	Control	*p*	Case	Control	*p*
	(N = 4107)	(N = 3710)		(N = 779)	(N = 667)	
Age, N (100%)			<0.001			0.358
≤60	2028 (49.38)	2013 (54.26)		294 (37.74)	268 (40.18)	
>60	2079 (50.62)	1697 (45.74)		485 (62.26)	399 (59.82)	
Gender, N (100%)						0.781
Male	0	0		511 (65.60)	443 (66.42)	
Female	4107 (100)	3710 (100)		268 (34.40)	224 (33.58)	
Smoking status, N (100%)						<0.001
Ever	0	0		421 (54.04)	297 (44.53)	
Never	4107 (100)	3710 (100)		358 (45.96)	370 (55.47)	
Histology type, N (100%)						
Squamous cell carcinoma	654 (15.92)			203 (26.06)		
Adenocarcinoma	3453 (84.08)			576 (73.94)		

**Table 2 cancers-14-05309-t002:** Details information of 28 candidate apaQTL-SNPs.

No.	Histology Type	SNP	Location	Gene	apaQTL-*p* Value	MAF (CHB)	MAF (CHS)	MAF (JPT)
1	LUAD	rs1130663	chr11:837582	*CD151*	8.11 × 10^−92^	0.1359	0.1048	0.0817
2	LUAD	rs4899152	chr14:65387116	*CHURC1*	1.75 × 10^−95^	0.1456	0.1333	0.1298
3	LUAD	rs10138506	chr14:65388243	*CHURC1*	3.33 × 10^−14^	0.0534	0.0333	0.0529
4	LUAD	rs3785	chr15:23005202	*NIPA2*	6.09 × 10^−15^	0.3058	0.4190	0.3654
5	LUAD	rs8069673	chr17:30661250	*C17orf75*	2.02 × 10^−14^	0.0340	0.0667	0.1587
6	LUAD	rs4802607	chr19:49959364	*ALDH16A1*	1.14 × 10^−13^	0.3883	0.4381	0.3029
7	LUAD	rs6151429	chr22:51063477	*ARSA*	1.90 × 10^−41^	0.0291	0.0238	0.0144
8	LUSC	rs17116169	chr5:153837482	*SAP30L*	6.23 × 10^−12^	0.0534	0.0524	0.0288
9	LUSC	rs2268314	chr7:44695725	*OGDH*	1.34 × 10^−13^	0.4320	0.4571	0.4571
10	LUSC	rs10999323	chr10:72174390	*EIF4EBP2*	4.04 × 10^−11^	0.3398	0.2714	0.4471
11	LUSC	rs1130698	chr11:838542	*CD151*	6.51 × 10^−80^	0.1019	0.0762	0..0529
12	LUSC	rs1130719	chr11:838760	*CD151*	2.02 × 10^−80^	0.1408	0.1048	0.0817
13	LUSC	rs7302556	chr12:66532242	*TMBIM4*	4.97 × 10^−31^	0.0194	0.0333	0.0192
14	LUSC	rs1615416	chr12:66549321	*TMBIM4*	2.23 × 10^−19^	0.1117	0.1095	0.0673
15	LUSC	rs11176067	chr12:66557162	*TMBIM4*	3.38 × 10^−107^	0.3641	0.3667	0.3462
16	LUSC	rs1185888	chr12:66560879	*TMBIM4*	9.61 × 10^−103^	0.1650	0.2048	0.2019
17	LUSC	rs169562	chr13:32998362	*N4BP2L2*	3.54 × 10^−16^	0.2718	0.3524	0.3558
18	LUSC	rs798272	chr13:33061719	*N4BP2L2*	5.48 × 10^−13^	0.4466	0.4429	0.4471
19	LUSC	rs45604	chr13:33099347	*N4BP2L2*	3.89 × 10^−13^	0.3301	0.3905	0.4087
20	LUSC	rs10138534	chr14:65387989	*CHURC1*	4.33 × 10^−134^	0.1456	0.1333	0.1298
21	LUSC	rs72726301	chr14:65395668	*CHURC1*	1.52 × 10^−36^	0.0922	0.1000	0.0769
22	LUSC	rs1064108	chr14:65400265	*CHURC1*	3.48 × 10^−61^	0.2670	0.2667	0.3654
23	LUSC	rs7224742	chr17:30657058	*C17orf75*	1.00 × 10^−15^	0.0340	0.0667	0.1587
24	LUSC	rs73572386	chr20:3849736	*MAVS*	1.25 × 10^−12^	0.2864	0.3048	0.2837
25	LUSC	rs8141941	chr22:19166263	*SLC25A1*	4.32 × 10^−15^	0.2573	0.2762	0.1779
26	LUSC	rs2481	chr22:36677400	*MYH9*	2.63 × 10^−15^	0.3786	0.3571	0.3462
27	LUSC	rs7073	chr22:43266363	*PACSIN2*	8.75 × 10^−12^	0.0728	0.0810	0.0433
28	LUSC	rs6151429	chr22:51063477	*ARSA*	7.04 × 10^−35^	0.0291	0.0238	0.0144

**Table 3 cancers-14-05309-t003:** Characteristics of the three SNPs in this study.

SNPs	Histology Type	Gene	Alleles	Cases	Controls	MAF (Cases)	MAF (Controls)	OR (95% CI) ^a^	*p* ^a^
rs10138506	LUAD	*CHURC1*	A > G	3048/390/15	3324/380/6	0.061	0.053	1.16(1.01–1.33)	0.034
	LUSC	*CHURC1*	A > G	596/55/3	3324/380/6	0.047	0.053	0.87(0.66–1.15)	0.325
rs1130698	LUAD	*CD151*	T > C	2729/675/49	2915/742/53	0.112	0.114	0.98(0.88–1.08)	0.687
	LUSC	*CD151*	T > C	493/149/12	2915/742/53	0.132	0.114	1.19(1.00–1.41)	0.049
rs1130719	LUAD	*CD151*	A > T	2560/824/69	2751/882/77	0.139	0.140	1.00(0.91–1.10)	0.990
	LUSC	*CD151*	A > T	455/175/24	2751/882/77	0.170	0.140	1.26(1.08–1.48)	0.002

^a^ Logistic regression analysis adjusted for age status in the additive model.

**Table 4 cancers-14-05309-t004:** Specific information of three candidate apaQTLs in the screening and validation phase.

Histology Type	SNPs	Phase	Genotypes	Cases, N (100%)	Controls, N (100%)	Adjusted OR (95% CI) ^a^	*p ^a^*
LUAD	rs10138506	Screening	AA	3048 (88.27)	3324 (89.60)	1 (ref)	--
	*CHURC1*		AG	390 (11.30)	380 (10.24)	1.12 (0.97–1.30)	0.133
			GG	15 (0.43)	6 (0.16)	2.72 (1.05–7.02)	0.038
			Dominant model			2.69 (1.04–6.94)	0.041
			Recessive model			1.14 (0.99–1.32)	0.071
			Additive model			1.16 (1.01–1.33)	0.034
		Validation	AA	489 (84.90)	591 (89.01)	1 (ref)	--
			AG	86 (14.93)	72 (10.84)	1.45 (1.03–2.03)	0.033
			GG	1 (0.17)	1 (0.15)	1.06 (0.07–17.00)	0.969
			Dominant model			1.44 (1.03–2.02)	0.034
			Additive model			1.42 (1.02–1.98)	0.038
LUSC	rs1130698	Screening	TT	493 (75.38)	2915 (78.57)	1 (ref)	--
	*CD151*		TC	149 (22.78)	742 (20.00)	1.19 (0.98–1.44)	0.087
			CC	12 (1.84)	53 (1.43)	1.39 (0.75–2.56)	0.295
			Dominant model			1.34 (0.73–2.46)	0.351
			Recessive model			1.20 (0.99–1.45)	0.060
			Additive model			1.19 (1.00–1.41)	0.049
		Validation	TT	153 (75.37)	535 (80.21)	1 (ref)	--
			TC	47 (23.15)	126 (18.89)	1.36 (0.91–2.02)	0.138
			CC	3 (1.48)	6 (0.90)	1.46 (0.33–6.48)	0.615
			Dominant model			1.36 (0.92–2.02)	0.125
			Additive model			1.32 (0.92–1.90)	0.130
LUSC	rs1130719	Screening	AA	455 (69.57)	2751 (74.15)	1 (ref)	--
	*CD151*		AT	175 (26.76)	882 (23.77)	1.20 (1.00–1.45)	0.052
			TT	24 (3.67)	77 (2.08)	1.86 (1.18–2.95)	0.008
			Dominant model			1.78 (1.13–2.81)	0.014
			Recessive model			1.25 (1.05–1.50)	0.012
			Additive model			1.26 (1.08–1.48)	0.002
		Validation	AA	147 (72.41)	482 (73.59)	1 (ref)	--
			AT	49 (24.14)	154 (23.51)	1.17 (0.79–1.73)	0.428
			TT	7 (3.45)	19 (2.90)	1.28 (0.50–3.26)	0.611
			Dominant model			1.18 (0.81–1.72)	0.378
			Additive model			1.15 (0.84–1.58)	0.372

^a^ Logistic regression analysis adjusted for age, gender and smoking status.

## Data Availability

The data that support the findings of this study are available from the corresponding author upon reasonable request.

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
