# Peer review of "A Novel apaQTL-SNP for the Modification of Non-Small-Cell Lung Cancer Susceptibility across Histological Subtypes"

_cancers, 2022, doi:10.3390/cancers14215309_

Round 1

Reviewer 1 Report

1) The main concern about this study is lack of evaluation of this product at the protein. It is strongly suggested that the authors widely discussed this issue. 

2) The role of clinical significance of these data should be clearly explained.

3) The possible correlation between these findings with stage and survival data should be added in the manuscript.  

4) The role of genetic background should be added in the discussion. 

Author Response

Dear reviewer,

      Attached please find the responses. 

Reviewer 2 Report

The authors have done a good job with this study. The following points to help provide more clarity for the reader

1. Introduction Line 43: Could the authors please add the global values?

2. Introduction line 45: Could the authors please indicate how the current study might help address this problem and reduce the incidence of NSCLC in the general population

3. Line 112: Could the authors indicate if the controls were only female? Could the authors also indicate why a female data set was selected for Stage I?

4. Line 115: Similarly, could the authors indicate the sex of the patients and healthy controls included in stage II of the study?

5. Figure 2: Please include a more detailed description of the 3 plots in the figure legend

6. Please include a detailed description of the figures in the legend as well.

7. Could the authors also tabulate the characteristics of the Stage II subjects in the first paragraph of the results section?

Author Response

(The authors gave the same response as above.)

Reviewer 3 Report

Dear authors,

Congratulations on your work and result.

Your article brings novelty for the molecular oncogenic mechanisms in NSCLC, which may become a biomarker for this disease.

The results are clearly presented, the materials and methods well described and the discussions relevant for the topic.

As a suggestion, you could replace the word „stage” with another synonym (phase?) because it can be mistaken by staging of cancer according to TNM.

The figures (particularly 2 and 4) could be a little wider because they are hard to „read”.

Moreover, the p-value in table 4 should not appear on 2 rows as it is unusual.

King regards,

Author Response

(The authors gave the same response as above.)

Round 2

Reviewer 1 Report

Accept.